# Influence of Extrudate-Based Textural Properties on Pellet Molding Quality

**DOI:** 10.3390/ph16101505

**Published:** 2023-10-23

**Authors:** Wenxiu Tian, Xue Li, Wenjie Li, Aile Xue, Minyue Zheng, Xiao Lin, Yanlong Hong

**Affiliations:** 1Shanghai Innovation Center of TCM Health Service, Shanghai University of Traditional Chinese Medicine, No. 1200, Cai-Lun Road, Pudong District, Shanghai 201203, China; 22021554@shutcm.edu.cn (W.T.); m15242054511@163.com (X.L.); liwenjie@shutcm.edu.cn (W.L.); 22022550@shutcm.edu.cn (A.X.); zmy2290677028@126.com (M.Z.); 2College of Chinese Materia Medica, Shanghai University of Traditional Chinese Medicine, No. 1200, Cai-Lun Road, Pudong District, Shanghai 201203, China

**Keywords:** pellets, extrudate, extrusion–spheronization, surface roughness, MCC, traditional Chinese medicine (TCM) extracts

## Abstract

As the precursor of pellets, the extrudate has a direct impact on the molding quality of the pellets. Therefore, the correlation between the surface roughness of the extrudates and the molding quality of pellets with pure microcrystalline cellulose (MCC) formulations and those containing traditional Chinese medicine (TCM) formulations was explored. MCC was used as a pelleting agent, mixer torque rheometry (MTR) was used to guide the optimal dosage of the wetting agent, and TCM extracts (drug loadings of 20% to 40%) were selected as model drugs to prepare the extrudates and pellets under the same extrusion spheronization process conditions. The surface roughness and texture parameters of extrudates were analyzed via a microscope and texture analyzer, respectively, and the quality of pellets was evaluated. The extrudate roughness of the pure MCC prescription decreased and then increased with increasing water addition, while the extrudate roughness of the prescription containing TCM extracts tended to increase and then decrease. The addition of water affected the extrudate properties, with TCM extract molecules filling gaps in the MCC structure, leading to rough surfaces. The extrudate roughness of the TCM prescriptions was significantly greater than that of the pure MCC prescriptions at optimal water addition levels, resulting in ideal pellets.

## 1. Introduction

Pellets are spherical or quasi-spherical oral solid preparations with a particle size of 0.5~2.5 mm that have the advantages of low local stimulation, high bioavailability, and preparation of multiple drug release systems [1]. The extrusion–spheronization method for preparing pellets has simple equipment and processing and good reproducibility of pellet quality, and it has become an internationally common preparation method [2].

When preparing pellets with pure microcrystalline cellulose (MCC) as the raw material, it is widely believed that the smoother the surface of the extrusion, the better the quality of the pellets prepared. The pellets prepared with a smooth extrusion surface have a high yield, good roundness, and a small particle size distribution [3,4,5,6,7,8,9]. Using different pellet preparation techniques also showed that the pellets obtained from extrudates with smooth surfaces had larger particle size, higher density, better fluidity, and characteristics of slow release and controlled release [10,11]. It was also found that when the ratio of screen diameter was ≤1, the surface of the extrudate was rough fish scale, the rounding result was powder, and the pellets had a large particle size distribution. When the ratio was large, the screen was thick, the surface of the extrudate was smooth, and pellets with high roundness and good fluidity were obtained via rolling [10]. Therefore, traditional Chinese medicine (TCM) scholars have studied and reported on the understanding that “the surface of the extrudate of ideal pellets should be smooth” [12,13,14,15,16] and optimized the formulation and process based on this deduction. Some scholars [17] reviewed the application of extrusion–spheronization TCM pellets and believed that the ideal extrudate should be smooth, dense, and uniform in color. According to some researchers [18], a certain amount of wetting agent should be added to the mixed powder of TCM extracts and excipients. After being pressurized in the extruder, the particles are rearranged to fill the gaps in the powder layer and tightly combined. However, the extrudate that is dense and hard cannot be easily broken and rounded. As a result, it tends to form short rods. In addition, during the preparation of the pellets containing *Gardeniae Fructus* extract [19], when the extrusion speed was too fast, the wet masses were rapidly extruded, resulting in loose extrudates and more fine powder after rounding. An interesting result was found in our preliminary study; for prescriptions containing single TCM extracts with drug loadings of 20–40%, extrudates with smooth surfaces failed to form spherical pellets, and the extrudates that could prepare ideal pellets had relatively rough surfaces. This result was the opposite of that reported in the literature using pure MCC as the model drug.

Raw materials are the material basis for the preparation of pellets via extrusion–spheronization. As the precursor of pellets, wet mass and extrudate have a direct impact on the molding quality of pellets. In practical applications, the extrudate is often studied in place of wet mass. The extrudate is the cylindrical particle or strip extrudate made of wet mass through a screen with a certain thickness and aperture under the continuous compression force of the extruder screw. Its surface characteristics (surface structure, surface roughness, etc.) and texture characteristics (adhesion, hardness, elasticity, resilience, cohesion and mastication, etc.) can directly affect the molding quality of pellets [20,21]. The extrusion speed affects the surface roughness of the extrudates. The two components of a self-emulsifying system were separated into pellets prepared via extrusion–spheronization, using MCC as a matrix excipient. If the extrusion was too fast, the surface of the extrudates was mostly rough or fish scale; thus, the extrudates were easily cut off, and the particle size was unequal during the rolling process, which ultimately led to an increase in the fine powder content, a decrease in the yield, and a wide particle size distribution range [22]. In addition, the moisture in the soft material could not lubricate the screen surface due to the high extrusion speed. As a result, the extrusion force was increased, the temperature inside the extruder continued to rise, the surface defects of the extruder were fish scale, and the pellet size and roundness were reduced [23]. The wetting agent affects the texture characteristics of the extrudates. Increasing the amount of wetting agent increases the plasticity and resilience of the extrudates, resulting in particle sizes of pellets that are too large or even unable to be rounded [24]. The addition of a low concentration of surfactant to the wetting agent affects the physical properties of the extrudates, such as cohesion, thus affecting the quality of the pellets [25]. Our research team used a texture analyzer to characterize the texture characteristics of the extrudates in the process of preparing pellets via extrusion–spheronization with multiple parameters, and the adhesion, hardness, elasticity, cohesion, resilience, and chewiness of the extrudates were simultaneously measured via texture profile analysis (TPA) [26]. Further research showed that the forming quality of pellets could be predicted by evaluating the above physical properties of wet mass.

In this study, prescriptions containing TCM extracts and those containing pure MCC were used as two types of model drugs. The surface roughness and textural properties of the extrudates were determined, and the molding qualities of the extrudates and pellets were classified. Based on this premise, the correlations between the surface properties of extrudates, textural properties of extrudates, and moldability of pellets were further established. Thus, the mechanism of the different correlations between extrudate surface properties and pellet molding quality for different formulation raw materials (containing TCM extracts or pure MCC) was explored.

## 2. Results

### 2.1. Correlation between Extrudate Surface Roughness and Pellet-Forming Quality When Pure MCC Is Used as the Pelleting Agent

The extrusion–spheronization results of 28 pure MCC formulations and the surface roughness (R) of the extrudates are shown in Table 1 and Figure 1 and Figure 2.

As shown in Table 1 and Figure 1, among the 28 MCC prescriptions, the extrudates of 10 prescriptions were categorized as E1 and could not be successfully extruded. Fourteen prescriptions were smoothly extruded with a smooth surface, and these resulted in round pellets. Three prescriptions were smoothly extruded and had a rough surface, two of which resulted in fine powders and one of which resulted in irregular lumps after rounding. The extrudates of one prescription were categorized as E4 and became the large ball after rounding. Therefore, the surface of the extrudate was smooth for the prescriptions that yielded round pure MCC pellets; however, when the surface of the extrudate was rough, round pure MCC pellets could not be prepared, which was consistent with reported results.

As shown in Figure 2, the roughness of extrudates of the E1 type was difficult to distinguish from those of the extrudates of the other types. This may be because there is less wetting agent to induce the viscosity of the MCC, the sample struggles to adhere to the screw, and the sample is difficult to extrude. Additionally, because the wet mass was loosely piled together in the extrusion process, the extrudates did not easily form strips when extruded. More heat was produced and more adhesive evaporated with the longer extrusion time, and irregular shapes were more easily formed. When there was too much water, under the action of extrusion pressure, the excessive water migrated to the surface of the extrudates, resulting in extrusion aggregation. A *t* test was used to compare the extrudates R of E2 and E3, and the difference was statistically significant (*p* < 0.01). When R was ≤1.20, the surfaces of the pure MCC pellet extrudates were smooth, and the sphericity results were S3. When R was >1.20, the extrudates of pure MCC pellets were rough, and the results of spheronization were S1 and S4.

In summary, for the pure MCC prescription, the extrusion surface was smooth, and the quality of the pellet was high.

### 2.2. Correlation between the Extrusion Roughness and Pellet-Forming Quality Using Single or Compound TCM Extracts as Model Drugs

The results of the extrusion–spheronization and the surface roughness of the extrudate for 300 prescriptions containing single TCM extracts are detailed in Appendix A and Figure 3 and Figure 4.

As shown in Appendix A and Figure 3, among the 300 prescriptions containing single TCM extracts, 160 prescriptions with spherical pellets were obtained, and the extrudates were all rough or scaly. Therefore, among the 25 kinds of single TCM extracts, the extrudates that could prepare ideal TCM pellets needed to have rough surfaces. There were 184 prescriptions that could be extruded successfully, and the surface of the extrudate was rough. In total, 10 dumbbells or double spheres, 160 spheres, and 14 mucilaginous or large spheres were obtained after rounding. This result indicated that in the 25 single TCM extract prescriptions studied, the extrudates were rough, and all did not have spherical particles. Out of 37 extrudates with smooth surfaces, dumbbells or double spheres were obtained after rounding, indicating that spherical particles could not be prepared when the extrudates were smooth. In summary, among the prescriptions containing single TCM extracts, the surfaces of the extrudates of the prescriptions for spherical pellets were rough or scaly. For the prescriptions that could be extruded successfully, the surfaces of the extrudates were rough, and the product obtained after rounding was dumbbell-shaped or bi-spherical, spherical, sticky-wall spherical, or large spherical. However, regarding the prescriptions that could be extruded successfully for which the surface of the extrudate was smooth, the product obtained after rounding was dumbbell-shaped or double-spherical particles.

As shown in Figure 4, E1 and E3 partially overlapped, which potentially occurred because when the amount of water added was too low, the materials were difficult to extrude, and these elements piled up together. Under extrusion pressure, a large amount of heat was generated, resulting in rough surface defects of the extrudates. In this experiment, under the premise that the extrudate could be extruded smoothly and was not aggregated, when R was ≤1.20, the surfaces of the extrudates containing single TCM extract prescription pellets were smooth, and the result was S2. When R was >1.2, the extrudates containing single TCM extract prescription pellets were rough, and the pellets were mostly S3.

These results indicated that it was difficult to prepare desirable pellets in prescriptions containing single TCM extracts when the extrudates were smooth.

The surface roughness values of the extrudates of the MCC prescription and three loaded single TCM extracts prescriptions at four water addition levels were averaged. A line graph was created in order to clearly show the difference between the two model drugs, as shown in Figure 5.

In Figure 5, the extrudate roughness of the pure MCC prescription tends to decrease and then increase with increasing water addition; in contrast, the extrudate roughness of the prescription containing single TCM extracts tends to increase and then decrease. More importantly, the extrudate roughness of the single TCM extract prescriptions was significantly greater than that of the pure MCC prescriptions at optimal water addition levels 2 and 3, which resulted in ideal pellets.

The results of the extrusion–spheronization and the surface roughness of the extrudates of the 72 prescriptions containing the compound TCM extracts are shown in Table 2 and Figure 6.

As shown in Table 2 and Figure 6, 42 of the 72 prescriptions containing compound TCM extracts were rolled to obtain spherical pellets, and the surfaces of the extrudates were rough or scaly. Therefore, among the six compound TCM extracts, the surface of the extrudates that could prepare ideal TCM pellets needed to be rough. The extrudates of 63 prescriptions could be smoothly extruded with rough surfaces. After the rounding process, 1 fine powder, 3 dumbbell or double spheres, 42 spherical particles, and 17 mucilaginous or large spheres were produced. The results indicated that in the six compound TCM extracts examined, the rough surfaces of all extrudates did not yield spherical pellets. In three prescriptions, the extrudates had smooth surfaces, and the pellets were dumbbell shaped or bi-spherical after rounding. Thus, in six compound TCM extracts, the extrudates with smooth surfaces were unable to form spherical pellets. In the prescriptions containing compound TCM extracts, the surfaces of the extrudates used to obtain ideal pellets were all rough; when the extrudates could be smoothly extruded and had rough surfaces, the four types of pellets were obtained, while when the extrudates could be smoothly extruded and had smooth surfaces, only rod-shaped or dumbbell-shaped pellets could be obtained.

Therefore, in the prescriptions containing compound TCM extracts, the surface of the extrudate was smooth, and the quality of the pellets was not high.

In summary, for the pure MCC prescriptions, a smoother surface of the extrudate correlated to a better quality of the pellet. For prescriptions containing TCM extracts, the smooth surface of the extrudate did not result in high-quality pellets. The surface of the extrudate for the preparation of spherical pellets needed to be rough, and when the surface of the extrudate was smooth, spherical pellets could not be prepared.

### 2.3. Structural Properties of the Extrudates

To investigate the reasons for the differences in the correlation between the surface properties of the two types of model drug extrudates and the formability of pellets, each qualitative parameter of the extrudates containing single TCM prescriptions and those containing pure MCC prescriptions were measured and analyzed together with the surface roughness of the extrudates using the method outlined below.

For 300 prescriptions containing single TCM extracts, the structural properties of extrudates were determined, as shown in Appendix A.

For the prescriptions containing single TCM extracts, the six qualitative parameters and surface roughness statistics of the four types of extrudates were tested for normality based on the physical properties and surface roughness results of the extrudates; these data did not follow a normal distribution. The homogeneity of variance test could not be performed. Therefore, the Kruskal–Wallis non-parametric test was used, and the original hypothesis was rejected at *p* < 0.01, indicating that there were significant differences in the textural properties of the four extrusions.

The textural properties of the extrudates for the 28 pure MCC prescriptions are listed in Table 3.

To explore the reasons for the difference in the correlation between the surface properties of the extrudate and the quality of the pellets in the two model drugs, a multiple comparison of the texture parameters and the surface roughness of the extrudates containing single TCM prescriptions and pure MCC prescription pellets with ideal results (i.e., rounded results) was carried out via least significant difference (LSD), and the results are shown in Table 4.

Based on the results from multiple comparisons of LSD, significant differences in Sp, Co, Ch, and R between the E3-S3 _TCM_ and E2-S3_MCC_ groups were observed. The R value of the E3-S3 _TCM_ group was higher than that of the E2-S3_MCC_ group, the Sp of the E3-S3 _TCM_ group was lower than that of the E2-S3_MCC_ group, and the Co of the E3-S3 _TCM_ group was higher than that of the E2-S3_MCC_ group. Therefore, the R value of surface roughness potentially had a positive correlation with Co.

To further explore the mechanism, Spearman’s correlation analysis was conducted on the textural parameters and surface roughness of the extrudate for the two types of model drugs, and the results are provided in Table 5 and Table 6. Based on Table 5, Ha, Ad, Sp, Co, and Re were significantly correlated with R. Among them, Ad and Sp were negatively correlated with R, while other texture parameters were positively correlated with R. The correlations were Ad, Co, Sp, Re, and Ha in descending order, and Ad, Co, and Sp had the greatest influence on the R value of the extrudate. Table 6 lists the results of Spearman’s correlation analysis for the pure MCC prescriptions: no significant correlation was observed between the texture parameters and surface roughness, potentially due to the different forming mechanisms of the pure MCC pellets and TCM pellets.

The qualitative parameters Ha, Ad, Sp, Co, Ch, and Re of the extrudates containing single herbal prescriptions were set as independent variables X1, X2, X3, X4, X5, and X6, respectively. R was set as the dependent variable Y. Stepwise regression analysis was performed, and the results of the analysis are shown in Table 7 and Table 8.

From Table 8, the stepwise regression equation was Y = 1.203 − 9.697 × 10^−5^ X1 − 0.104 X3 + 0.829 X4. From the analysis of variance table (ANOVA), F = 24.356, *p* = 0.000, and the regression equation had statistical significance. From the T test of the regression coefficient, variables for which *p* < 0.05 had statistical significance. The standardized regression coefficient with the order of influence on Y was X4 > X3 > X1. Therefore, the influence on surface roughness R (Y) from highest to lowest was Co (X4), Sp (X3), Ha (X1), and Co, where Co had the greatest influence.

Based on the analysis results, the preliminary conclusion was drawn that extrudate surface roughness was related to parameters such as Co, Ad, Ha, Re, Sp, etc. Among them, the correlation between Co, Ad, Sp, and extrudate surface roughness was the largest, and the R value of extrudate surface roughness was positively correlated with Co and negatively correlated with Ad and Sp.

Therefore, the main factor leading to the difference in the surface characteristics (surface roughness) of the extrudates containing the single TCM prescription and pure MCC prescription was the inconsistency of the texture parameter Co in the two kinds of prescriptions. Co was the adhesion force between molecules inside the extrudates; specifically, this referred to the cohesiveness of the extrudates pulled together. A greater Co value correlated to a smaller compressibility. The sudden increase in Ad and Sp in the later stages of water addition was an important reason why the extrudates containing herbal extracts did not produce the desired pellets during the preparation process.

## 3. Discussion

Raw material is the material basis for the preparation of pellets via the extrusion rounding method, and the extrudate, as the precursor of pellets, has a direct impact on the molding quality of pellets. The literature research [3,4,5,6,7,8,9,10,11] shows that the smoother the surface of the extrudate, the better the molding quality of the pellet, which is an important guiding significance for the preparation of pellets via the extrusion and rounding method. Researchers in traditional Chinese medicine basically follow the above theory when conducting studies of pellets found in traditional Chinese medicine [12,13,14,15,16,17,18,19]. However, there are no definitive studies stating that the theory is suitable for model drugs such as traditional Chinese medicine extracts. Therefore, in this study, the correlation law between the surface properties of extrudates and the molding quality of pellets was investigated. As a result of our study, it was found that for two different model drugs, namely pure MCC and those containing herbal raw materials, the surface properties of the extrudates showed different correlations with the quality of pellet molding.

The reasons for this difference may be that with the increase in the amount of water and high-speed mixing, the wetting agent, i.e., water, will be between the particles in four states [27]. The first is the migration of water molecules between the particles, showing a pendulum phase (single bridge); further saturation will produce inter-particle forces, altering the performance of the funicular (single bridge partially filled), and then become suitable for the granulation point of the state, i.e., the capillary state [28] (multi-bridges are almost completely filled). There is excessive wetting when further water is added, and it takes on a continuous droplet phase.

For the pure MCC prescription, the R of the extrudate tended to decrease and then increase with increasing water addition in the upper and lower ranges of the wetting agent dosage for the preparation of the spherical pellets. When the amount of wetting agent did not reach the amount required for the preparation of spherical pellets, because the pure MCC small molecules are spatial network structures, a small amount of wetting agent could not induce its viscosity. As a result, water existed in a pendulum shape in the particle gap, part of the particle gap was not filled, and the adhesion force was small. It is difficult for the material to adhere to the screw and be extruded smoothly. In addition, because the material gathered at the screen mouth was subjected to greater friction and generated much heat, some of the water was evaporated. As a result, the extruded surface had more defects and tended to become irregular with a slightly rough surface. When the amount of wetting agent reached the level that could prepare spherical pellets, the liquid further filled the space between the particles and formed a cable band, and the adhesion force increased. Therefore, the material could adhere to the screw and be smoothly extruded, and the surface of the extrudate was smooth. When the water filling the space inside the particles was caused by capillary action, the water between the particles was saturated, the adhesive force between the molecules inside the extrudate was smaller, the deformation ability of the material was enhanced, the compressibility was good, the friction force was smaller during extrusion, and the surface of the extrudate was smooth. When the amount of wetting agent exceeded the appropriate amount of water required by the ideal pellet, the void inside the particle had a continuous droplet shape. The surface roughness of the extrudate increased relatively when the adhesion of the material was further increased to a certain level between the material and the screen surface.

For the prescription containing TCM extracts, the R of the extrude showed a trend of initially increasing and then decreasing within the upper and lower range of the amount of wetting agent used to prepare spherical pellets. When the amount of wetting agent did not reach the number of spherical pellets that could be prepared, the adhesion force was small, and the sample had difficulty adhering to the screw and being extruded. The interparticle bonding force was weak, and most of the extrudate had a slightly rough surface. When the amount of wetting agent reached the number of spherical pellets, the small molecules of the TCM dissolved in water and entered the internal space of the MCC to fill some of the gaps in the mesh structure. When the extrudate was extruded by the extrusion pressure of the screw, the internal cohesion Co between molecules was larger, and the compressibility was worse than that of pure MCC; thus, the surface roughness of the extrudate increased. When the amount of wetting agent exceeded the appropriate amount, the surface adhesion Ad and deformability Sp rapidly increased, and the roughness of the extrudate decreased. The extrudates bound to each other and formed irregular clusters or large balls during the preparation of the pellets; this was potentially an important reason for the failure of the prescriptions containing TCM extracts to produce ideal pellets at the later stage of water addition.

In summary, the surface roughness of extrudate was the result of co-action of Co, Ad, Ha, Re, and Sp. For the pure MCC prescription, a smoother extrudate surface correlated to a higher pellet-forming quality. For the prescription containing TCM extract, the extruded surface was smooth, and the pellet-forming quality was not high. The main factor affecting the surface roughness difference between the two-model drug extrudates was Co. The increase in the amount of water added with Ad and Sp at the later stage was an important reason that extrudates containing the extracts of Chinese medicine did not produce ideal pellets in the rolling process. This theory can enrich the molding mechanism of Chinese medicine pills prepared via the extrusion rounding method and provide new research ideas for the prescription and process design of Chinese medicine pellets.

## 4. Materials and Methods

### 4.1. Materials

The following equipment were used throughout our study: a particle size and shape analyzer (Belgium Occhio Instrument Company, Brussels, Belgium; model: Occhio500nano; software: Callisto; version number: 3.4); a microscope (AMG Company, Mill Creek, Washington, DC, USA, model: AME-3302); an electric thermostatic blast drying oven (Shanghai Jinghong Experimental Equipment Co., Ltd., Shanghai, China); a physical property tester (Stable Micro Systems, Godalming, UK); an A/BE probe (55 mm × 70 mm acrylic cup, 45 mm diameter disc); an extrusion–spheronization machine (Chongqing Yingge Pelletizing and Coating Technology Co., Ltd., Chongqing, China; model: E50-S250); a torque rheometer (Caleva, Dorset, UK, model: MITERTORQUEREHOMETER3); a portable high-speed universal crusher (Wenling Linda Machinery Co., Ltd., Zhejiang, China; model: DFT-250); and an electronic balance (Shanghai Jingtian Electronic Instrument Co., Ltd., Shanghai, China; model: JA5002).

All single TCM extracts were extracted via water, dried via spray, and sieved through 80 mesh. A total of 25 kinds of Chinese herbal extracts were acquired from China Resources Sanjiu Pharmaceutical Co., Ltd.; these were as follows: the extract of *Scutellariae Radix* (Batch No.:150501); the extract of *Chrysanthemi Indici Flos* (Batch No.:1509003); the extract of *Eriobotryae Folium* (Batch No.:150803); the extract of *Rhei Radix Et Rhizoma* (Batch No.:1509001); the extract of *Farfarae Flos* (Batch No.:201501); the extract of *Bupleuri Radix* (Batch No.:150902); the extract of *Schisandrae Chinensis Fructus* (Batch No.:150601); the extract of *Cynanchi Stauntonii Rhizoma Et Radix* (Batch No.:120409); the extract of *Acori Tatarinowii Rhizoma* (Batch No.:120409); the extract of *Cnidii Fructus* (Batch No.:120409); the extract of *Arecae Semen* (Batch No.:120409); the extract of *Herba Patriniae* (Batch No.:110901); the extract of *Sargentodoxae Caulis* (Batch No.:110802); the extract of *Magnoliae Flos* (Batch No.:120409); the extract of *Gardeniae Fructus* (Batch No.:201411); the extract of *Atractylodis Macrocephalae Rhizoma* (Batch No.:150602); the extract of *Citri Reticulatae Pericarpium* (Batch No.:150409); the extract of *Artemisiae Argyi Folium* (Batch No.:150604); the extract of *Sophorae Flavescentis Radix* (Batch No.:1508002); the extract of *Eucommiae Cortex* (Batch No.:150801); the extract of *Astragali Radix* (Batch No.:201501); the extract of *Leonuri Herba* (Batch No.:201504); the extract of *Siegesbeckiae Herba* (Batch No.:1509002); the extract of *Zingiberis Rhizoma* (Batch No.:120409); and the extract of *Gastrodiae Rhizoma* (Batch No.:20130404).

The extracts of compound Chinese medicine were laboratory-made samples derived from a laboratory-based project, and there were 6 kinds: Huoling Shengji Extract (Batch No.: 15060301); Xiao’er Qihua Extract (Batch No.: 20151010); Gouqi Zhichan Extract (Batch No.: 201409-1); Shexiang Baoxin Extract (Batch No.: 110428); Zhuju Hewei Extract (Batch No.: 20151221); and Kangerling Extract (Batch No.: Z170901).

There were 7 kinds of MCC with different manufacturers, models, and batch numbers; these were as follows: SH 101 (Manufacturer: Anhui Shanhe Pharmaceutical Auxiliary Materials Co., Ltd., Anhui, China; Batch No.: 160502); Avicel PH 101 (Manufacturer: FMC company, Philadelphia, PA, USA; Batch No.: P115828088); MT PH 101 (Manufacturer: Mingtai Chemical Co., Ltd., Taiwan, China; Batch No.: C1508003-S); Original PH 101 (Manufacturer: Shanghai Changwei Pharmaceutical Auxiliary Materials Technology Co., Ltd., Shanghai, China; Batch No.: P0101F510); JRS PH 101 (Manufacturer: JRS company, Weissenborn, Germany; Batch No.: 1016610141712); JRS PH 102 (Manufacturer: JRS company, Weissenborn, Germany; Batch No.: 5610261409); and Avicel PH 113 (Manufacturer: FMC company, Philadelphia, PA, USA; Batch No.: 40927C).

### 4.2. Methods

#### 4.2.1. Determination of Mixer Torque Rheometry (MTR)

The moto speed of MTR was set to 50. Under this condition, the average torque force was initially calibrated via idling for 50 s. Then, the raw materials were placed into the mixing tank of the torque rheometer and mixed at a constant speed for 30 s, and the average torque force was recorded for 20 s. During the rotation of the MTR propeller, after a certain amount of wetting, the agent was added according to the instrument prompts, the samples were mixed for 30 s, and then the average torque force was recorded for 20 s. The wetting agent was, in turn, added once every 50 s to circulate. The amounts and times of the additions were set in advance. The torque force corresponding to a 20–30 water dosage was collected to obtain the torque rheological curve. The corresponding maximum torque T_max_ and the corresponding amount of water-added W_Tmax_ was obtained from the curve. The amount of water added was used as a reference to determine the best amount of wetting agent for preparing wet mass [29].

For the prescription containing TCM extracts, MCC (SH-101) was used as the pelleting agent, and single Chinese medicine and compound Chinese medicine extracts were used as the model drugs. Twenty grams of the mixture of MCC with different drug loading ratios and single TCM and compound TCM extracts was weighed. There were two peaks, namely T_max-1_ and T_max-2_, and the corresponding wetting agent dosages were W_Tmax-1_ and W_Tmax-2_ in the torque rheological curve. The amount of wetting agent W_Tmax-2_ corresponding to T_max-2_ was used to optimize the amount of wetting agent in the preparation of TCM pellets via extrusion–spheronization. The pure MCC was weighed at 15 g, the torque rheological curve had only a peak value T_max_, and the corresponding wetting agent dosage was W_Tmax_; this was the optimal wetting agent dosage used to guide the preparation of pellets via extrusion–spheronization.

#### 4.2.2. Preparation and Classification of Extrudates

The prescriptions containing single herb and compound herb extracts were as follows: 25 single herb extracts and 6 compound herb extracts, with MCC (SH-101) as the pelleting agent; drug loadings of 20%, 30%, and 40%; and distilled water as the wetting agent. There were four levels of water added as the dosage of the wetting agent: two levels over the optimal dosage of the wetting agent guided via MTR, one level under the optimal dosage, and one level at the optimal dosage. There were 300 prescriptions for a single TCM extract (25 × 3 × 4) and 72 prescriptions for a compound TCM extract (6 × 3 × 4). Seven kinds of MCC from different manufacturers were used as model drugs, and distilled water was used as the wetting agent. Overall, 4 water addition levels were used as the dosage of the wetting agent, totaling (7 × 4) 28 formulation prescriptions.

The above formulas were used to prepare the extrudates and products under the same extrusion–spheronization conditions. The extrusion conditions were as follows: the mesh diameter was 0.6 mm, and the extrusion speed was 60 r/min. The prepared extrudates were divided into four categories according to the smoothness of the extrusion process and the surface roughness of the extrudates, as shown below. E1: extrudate cannot be extruded smoothly, and the total amount of extrudate obtained is less than half of the amount fed. E2: extrudate can be extruded smoothly with a smooth surface. E3: extrudate can be extruded smoothly with a rough surface. E4: extrudate can be extruded smoothly, but it is clumped together. Some of the prepared extrudates were used for determining their texture characteristics. Some extrudates were placed in a rounding machine for rounding to prepare the pellets. Another portion was placed in an air blast oven to dry at 60 °C for 3 h, and their roughnesses were measured.

#### 4.2.3. Preparation and Classification of the Pellets

A portion of the extruded products was placed into the extrusion–spheronization machine to prepare pellets. The spheronization parameters were as follows: the spheronization speed was 1500 r/min, and the spheronization time was 3 min. Next, the spheronization products were placed in an air blast drying oven to dry at 60 °C for 3 h. Then, 50 final products were randomly selected and classified according to their particle shape. When the number of particles in a certain shape exceeded 50%, the spheronization products were classified into this type. The main classification standards were as follows: S1: the extrudate after rounding becomes fine powder. S2: the extrudate is rounded to obtain pellets in the form of rods, dumbbells or double balls. S3: the extrudate is rounded to obtain pellets in the form of spheres. S4: the extrudate after rounding becomes a large ball or an irregularly shaped mass.

#### 4.2.4. Determination of the Surface Roughness of the Extrudates

Five dried extrudates were randomly selected from each prescription, cut into two relatively flat samples with a length of approximately 2 mm, and placed on the glass slide of the microscope (eyepiece 10×, objective lens 10×). A light source with 75% brightness was selected, and the position of the glass slide was adjusted such that the entire extrudate could be clearly photographed out of the current field of vision. One extrudate was photographed each time, and the pictures were analyzed via particle size analyzer software to calculate the Perim. D, Ell. L, and Ell. W of the particles. The roughness characterization method based on the area equivalent elliptical moment was selected; specifically, roughness(R) = (Perim. D × 3.14)/(2 × Ell. L + 2 × Ell. W) = 1.57 × Perim. D/(Ell. L + Ell. W). This formula was used to calculate the R value, and it indicated the degree of non-smoothness of the extrudate surface [30]. The R value for each prescription extrusion was the average of five replicate samples. A larger R value correlated to the greater surface roughness of the extrusion.

#### 4.2.5. Characterization of the Structural Properties of the Extrudates

A physical property tester was used to conduct multiparameter characterization of the structural properties of the wet mass, and the TPA method [31] was used to test the various physical properties of the extrudates, such as adhesion, hardness, elasticity, cohesion, resilience, and chewability. The probe was installed on the test arm of the physical property tester. Firstly, the sample cup was placed directly below the probe to calibrate the height of the probe; specifically, the probe was returned and increased to 8 cm from the bottom of the cup after the sample was tested in the cup. The extrudates from the wet mass prepared with different formulas were used as samples. A total of 30 g of sample was weighed each time and placed in the sample cup. The TPA procedure was performed to determine the physical properties. The average value was obtained through three parallel experiments. The following parameters were monitors and defined: hardness (Ha) (g) was the resistance of the extrusion to external forces; adhesiveness (Ad) (g.s) was the negative area of the curve from the first compression force of zero to the second compression and represented the size of the adhesion between particles and particle surfaces, including the adhesion between the same particle surface and different particle surfaces; springiness (Sp) was the ability of the extrudate to finally recover when the applied force disappeared; cohesiveness (Co) was the mutual attraction between the same particles in the same substance; chewiness (Ch) represented the ease of the extrudate being cut off; and resilience (Re) was the instant reversibility of the extrusion.

## 5. Conclusions

In this study, representative TCM extracts were selected as model drugs to investigate the correlation between the surface properties of their extrudates and the molding quality of pellets, as well as to explore this mechanism. More importantly, it was found that the correlation between the surface properties of extrudates and the molding quality of pellets varies among different raw materials (TCM prescription, pure MCC prescription). In pure MCC formulations, the spheronization results with smooth extruded surfaces were all spherical. In contrast, in formulations containing herbal extracts, the spheronization results with rough extrusion surfaces were almost always spherical. The results of the textural analysis indicated that the extrudates of the two model drugs had different structural characteristics under the conditions of ideal particle preparation using wetting agents. At the same time, the suitable range of extrudate surface roughness for the preparation of spherical TCM pellets was determined. Furthermore, it was found that the suitable range of surface roughness of the extrudate for the preparation of spherical herbal pellets is that the R-value should be more than 1.2.

Overall, this study explores “the correlation between the surface properties of extrudates and the molding quality of TCM pellets prepared by extrusion rounding method” for the first time and corrects the misconception that the research on the molding of TCM pellets is copied from the literature. Furthermore, it enriches the molding mechanism of TCM pellets prepared via the extrusion rounding method and provides new research ideas for the prescription and process design of TCM pellets.

## Figures and Tables

**Figure 1 pharmaceuticals-16-01505-f001:**
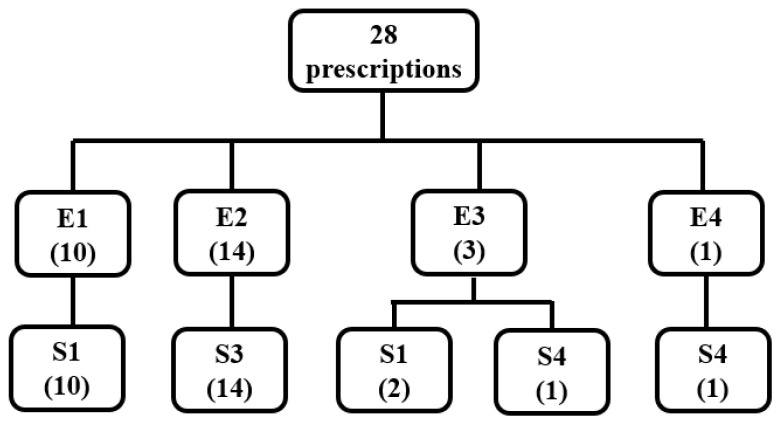
Classification of the extrusion–spheronization results for 28 prescriptions containing pure MCC (Notes—E1: extrudate cannot be extruded smoothly, and the total amount of extrudate obtained is less than half of the amount fed. E2: extrudate can be extruded smoothly with smooth surface. E3: extrudate can be extruded smoothly with a rough surface. E4: extrudate can be extruded smoothly, but it is clumped together. S1: the extrudate after rounding becomes a fine powder. S3: the extrudate is rounded to obtain pellets in the form of spheres. S4: the extrudate after rounding becomes a large ball or an irregularly shaped mass).

**Figure 2 pharmaceuticals-16-01505-f002:**
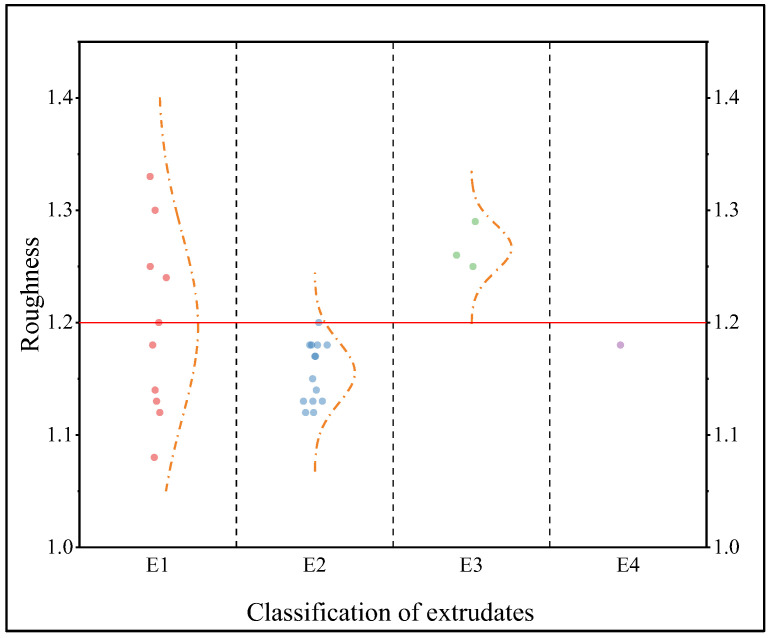
Scattering diagram of R for the four extrusions of the pure MCC prescription. (Notes—Roughness: surface roughness of the extrudates. E1: extrudate cannot be extruded smoothly, and the total amount of extrudate obtained is less than half of the amount fed. E2: extrudate can be extruded smoothly with a smooth surface. E3: extrudate can be extruded smoothly with rough surface. E4: extrudate can be extruded smoothly, but it is clumped together).

**Figure 3 pharmaceuticals-16-01505-f003:**
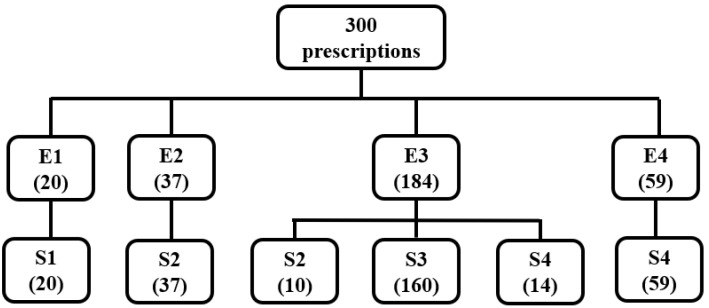
Classification of the extrusion–spheronization results for 300 prescriptions containing single TCM extracts. (Notes—E1: extrudate cannot be extruded smoothly, and the total amount of extrudate obtained is less than half of the amount fed. E2: extrudate can be extruded smoothly with a smooth surface. E3: extrudate can be extruded smoothly with a rough surface. E4: extrudate can be extruded smoothly, but it is clumped together. S1: the extrudate after rounding becomes a fine powder. S2: the extrudate is rounded to obtain pellets in the form of rods, dumbbells, or double balls. S3: the extrudate is rounded to obtain pellets in the form of spheres. S4: the extrudate after rounding becomes a large ball or an irregularly shaped mass).

**Figure 4 pharmaceuticals-16-01505-f004:**
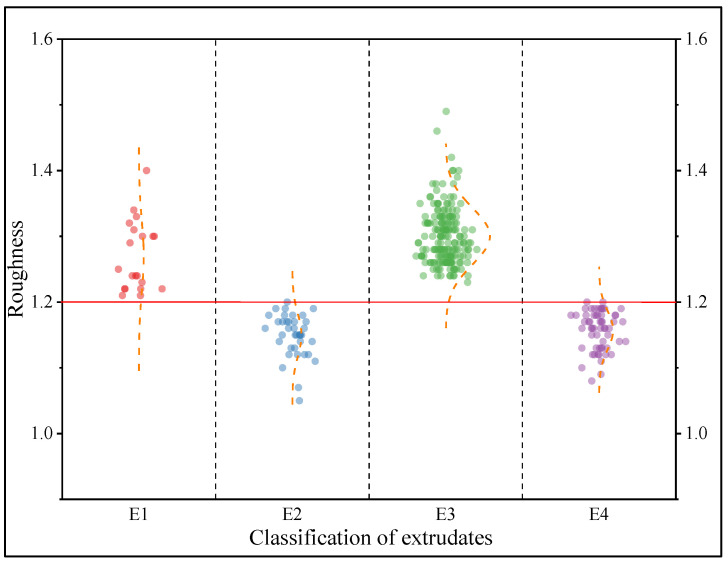
Scatter plot of R for the four extrusions of 300 prescriptions containing single TCM extracts (Notes—Roughness: surface roughness of the extrudates. E1: extrudate cannot be extruded smoothly, and the total amount of extrudate obtained is less than half of the amount fed. E2: extrudate can be extruded smoothly with a smooth surface. E3: extrudate can be extruded smoothly with rough surface. E4: extrudate can be extruded smoothly, but it is clumped together).

**Figure 5 pharmaceuticals-16-01505-f005:**
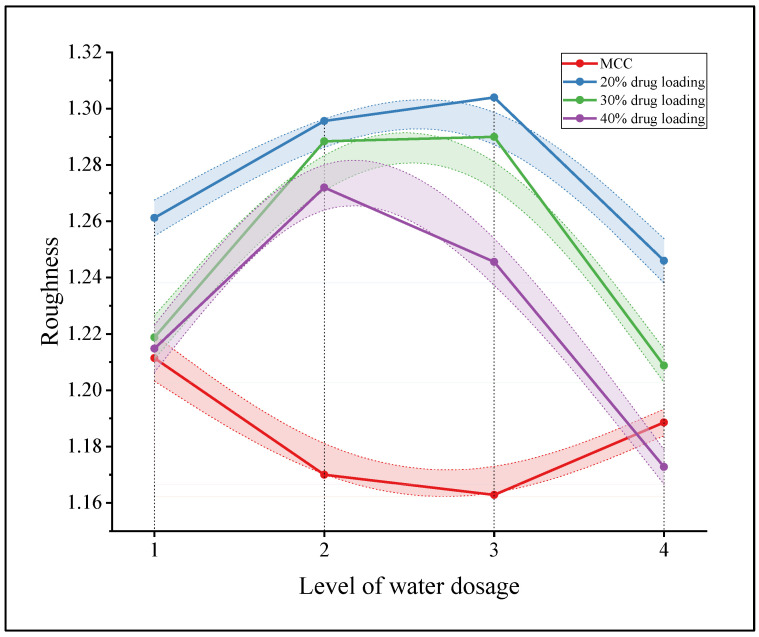
Variation in the extrudate roughness with the level of water addition for the pure MCC prescription and prescription containing single TCM extracts.

**Figure 6 pharmaceuticals-16-01505-f006:**
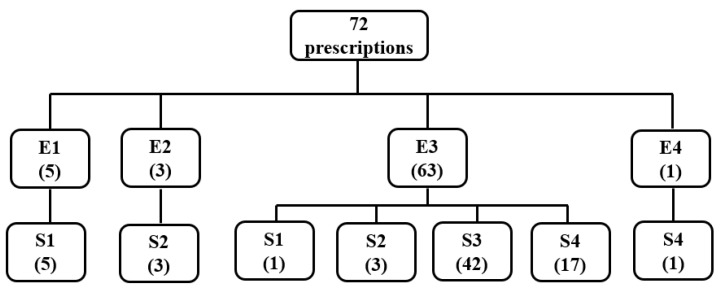
Classification of the extrusion and round results of 72 prescriptions containing compound TCM extracts. (Notes—E1: extrudate cannot be extruded smoothly, and the total amount of extrudate obtained is less than half of the amount fed. E2: extrudate can be extruded smoothly with a smooth surface. E3: extrudate can be extruded smoothly with a rough surface. E4: extrudate can be extruded smoothly, but it is clumped together. S1: The extrudate after rounding becomes a fine powder. S2: the extrudate is rounded to obtain pellets in the form of rods, dumbbells, or double balls. S3: the extrudate is rounded to obtain pellets in the form of spheres. S4: the extrudate after rounding becomes a large ball or an irregularly shaped mass).

**Table 1 pharmaceuticals-16-01505-t001:** R values and extrusion–spheronization results of pure MCC prescriptions (*n* = 5, x ± s).

No.	Model Drug	Water Added (%)	R	Extrusion Results	Rounding Results
1	Oricial101	100	1.20 ± 0.08	E1	S1
2	Oricial101	110	1.12 ± 0.06	E1	S1
3	Oricial101	120	1.12 ± 0.06	E2	S3
4	Oricial101	130	1.20 ± 0.04	E2	S3
5	JRS102	100	1.08 ± 0.07	E1	S1
6	JRS102	110	1.17 ± 0.06	E2	S3
7	JRS102	120	1.13 ± 0.03	E2	S3
8	JRS102	130	1.18 ± 0.03	E2	S3
9	Avicel113	100	1.14 ± 0.08	E1	S1
10	Avicel113	110	1.25 ± 0.02	E3	S1
11	Avicel113	120	1.13 ± 0.05	E2	S3
12	Avicel113	130	1.18 ± 0.05	E2	S3
13	Avicel101	100	1.18 ± 0.04	E1	S1
14	Avicel101	110	1.13 ± 0.06	E1	S1
15	Avicel101	120	1.14 ± 0.04	E2	S3
16	Avicel101	130	1.12 ± 0.03	E2	S3
17	MT101	100	1.33 ± 0.01	E1	S1
18	MT101	110	1.13 ± 0.07	E2	S3
19	MT101	120	1.18 ± 0.03	E2	S3
20	MT101	130	1.29 ± 0.01	E3	S4
21	JRS101	100	1.30 ± 0.11	E1	S1
22	JRS101	110	1.15 ± 0.07	E2	S3
23	JRS101	120	1.18 ± 0.04	E2	S3
24	JRS101	130	1.18 ± 0.05	E4	S4
25	SH101	100	1.25 ± 0.02	E1	S1
26	SH101	110	1.24 ± 0.03	E1	S1
27	SH101	120	1.26 ± 0.07	E3	S1
28	SH101	130	1.17 ± 0.05	E2	S3

Notes—R: surface roughness of the extrudates. E1: extrudate cannot be extruded smoothly, and the total amount of extrudate obtained is less than half of the amount fed. E2: extrudate can be extruded smoothly with smooth surface. E3: extrudate can be extruded smoothly with a rough surface. E4: extrudate can be extruded smoothly, but it is clumped together. S1: the extrudate after rounding becomes fine powder. S3: the extrudate is rounded to obtain pellets in the form of spheres. S4: the extrudate after rounding becomes a large ball or an irregularly shaped mass.

**Table 2 pharmaceuticals-16-01505-t002:** R-values and the extrusion–spheronization results for 72 prescriptions containing compound TCM extracts (*n* = 5, x ± s).

No.	Model Drug	Drug Loading (%)	Water Added (%)	$\bar{R}$	Extrusion Results	Rounding Results
1	Huoling Shengji Extract	20	70	1.21 ± 0.10	E1	S1
2	Huoling Shengji Extract	20	80	1.20 ± 0.03	E3	S3
3	Huoling Shengji Extract	20	85	1.22 ± 0.05	E3	S3
4	Huoling Shengji Extract	20	90	1.23 ± 0.05	E3	S4
5	Huoling Shengji Extract	30	50	1.24 ± 0.06	E3	S3
6	Huoling Shengji Extract	30	60	1.20 ± 0.01	E3	S3
7	Huoling Shengji Extract	30	65	1.24 ± 0.07	E3	S4
8	Huoling Shengji Extract	30	70	1.26 ± 0.02	E3	S4
9	Huoling Shengji Extract	40	25	1.16 ± 0.01	E3	S2
10	Huoling Shengji Extract	40	30	1.23 ± 0.06	E3	S3
11	Huoling Shengji Extract	40	40	1.25 ± 0.05	E3	S3
12	Huoling Shengji Extract	40	50	1.25 ± 0.02	E3	S4
13	Xiao’er Qihua Extract	20	70	1.13 ± 0.07	E1	S1
14	Xiao’er Qihua Extract	20	75	1.18 ± 0.04	E3	S3
15	Xiao’er Qihua Extract	20	80	1.23 ± 0.11	E3	S3
16	Xiao’er Qihua Extract	20	90	1.13 ± 0.06	E4	S4
17	Xiao’er Qihua Extract	30	40	1.19 ± 0.02	E3	S3
18	Xiao’er Qihua Extract	30	45	1.17 ± 0.06	E3	S3
19	Xiao’er Qihua Extract	30	50	1.26 ± 0.28	E3	S4
20	Xiao’er Qihua Extract	30	60	1.24 ± 0.13	E3	S4
21	Xiao’er Qihua Extract	40	30	1.18 ± 0.04	E2	S2
22	Xiao’er Qihua Extract	40	40	1.25 ± 0.07	E3	S3
23	Xiao’er Qihua Extract	40	50	1.29 ± 0.04	E3	S3
24	Xiao’er Qihua Extract	40	55	1.27 ± 0.09	E3	S4
25	Gouqi Zhichan Extract	20	60	1.21 ± 0.05	E3	S3
26	Gouqi Zhichan Extract	20	70	1.24 ± 0.06	E3	S3
27	Gouqi Zhichan Extract	20	80	1.19 ± 0.01	E3	S3
28	Gouqi Zhichan Extract	20	90	1.28 ± 0.07	E3	S4
29	Gouqi Zhichan Extract	30	40	1.21 ± 0.07	E3	S2
30	Gouqi Zhichan Extract	30	50	1.28 ± 0.07	E3	S3
31	Gouqi Zhichan Extract	30	55	1.22 ± 0.02	E3	S3
32	Gouqi Zhichan Extract	30	60	1.20 ± 0.05	E3	S3
33	Gouqi Zhichan Extract	40	30	1.18 ± 0.03	E2	S2
34	Gouqi Zhichan Extract	40	40	1.37 ± 0.16	E3	S3
35	Gouqi Zhichan Extract	40	50	1.27 ± 0.09	E3	S3
36	Gouqi Zhichan Extract	40	55	1.31 ± 0.09	E3	S4
37	Shexiang Baoxin Extract	20	70	1.25 ± 0.03	E1	S1
38	Shexiang Baoxin Extract	20	80	1.22 ± 0.01	E1	S1
39	Shexiang Baoxin Extract	20	90	1.26 ± 0.09	E3	S3
40	Shexiang Baoxin Extract	20	100	1.23 ± 0.10	E3	S3
41	Shexiang Baoxin Extract	30	70	1.20 ± 0.03	E1	S1
42	Shexiang Baoxin Extract	30	80	1.18 ± 0.04	E3	S3
43	Shexiang Baoxin Extract	30	90	1.22 ± 0.02	E3	S3
44	Shexiang Baoxin Extract	30	95	1.15 ± 0.03	E3	S4
45	Shexiang Baoxin Extract	40	60	1.21 ± 0.11	E3	S3
46	Shexiang Baoxin Extract	40	70	1.22 ± 0.00	E3	S3
47	Shexiang Baoxin Extract	40	80	1.20 ± 0.04	E3	S3
48	Shexiang Baoxin Extract	40	85	1.20 ± 0.06	E3	S4
49	Zhuju Hewei Extract	20	65	1.26 ± 0.01	E3	S3
50	Zhuju Hewei Extract	20	70	1.24 ± 0.05	E3	S3
51	Zhuju Hewei Extract	20	80	1.25 ± 0.01	E3	S3
52	Zhuju Hewei Extract	20	90	1.21 ± 0.08	E3	S4
53	Zhuju Hewei Extract	30	40	1.20 ± 0.05	E3	S2
54	Zhuju Hewei Extract	30	50	1.26 ± 0.01	E3	S3
55	Zhuju Hewei Extract	30	60	1.27 ± 0.07	E3	S3
56	Zhuju Hewei Extract	30	70	1.25 ± 0.06	E3	S4
57	Zhuju Hewei Extract	40	35	1.20 ± 0.06	E2	S2
58	Zhuju Hewei Extract	40	40	1.29 ± 0.04	E3	S3
59	Zhuju Hewei Extract	40	45	1.24 ± 0.02	E3	S3
60	Zhuju Hewei Extract	40	50	1.31 ± 0.09	E3	S4
61	Kanger Ling Extract	20	75	1.29 ± 0.08	E3	S3
62	Kanger Ling Extract	20	80	1.24 ± 0.01	E3	S3
63	Kanger Ling Extract	20	85	1.34 ± 0.03	E3	S3
64	Kanger Ling Extract	20	90	1.27 ± 0.01	E3	S4
65	Kanger Ling Extract	30	50	1.28 ± 0.03	E3	S1
66	Kanger Ling Extract	30	55	1.26 ± 0.04	E3	S3
67	Kanger Ling Extract	30	65	1.23 ± 0.03	E3	S3
68	Kanger Ling Extract	30	70	1.23 ± 0.04	E3	S4
69	Kanger Ling Extract	40	35	1.24 ± 0.02	E3	S3
70	Kanger Ling Extract	40	40	1.27 ± 0.07	E3	S3
71	Kanger Ling Extract	40	45	1.35 ± 0.05	E3	S3
72	Kanger Ling Extract	40	50	1.26 ± 0.00	E3	S4

Notes—R: surface roughness of the extrudates. E1: extrudate cannot be extruded smoothly, and the total amount of extrudate obtained is less than half of the amount fed. E2: extrudate can be extruded smoothly with a smooth surface. E3: extrudate can be extruded smoothly with a rough surface. E4: extrudate can be extruded smoothly, But it is clumped together. S1: the extrudate after rounding becomes a fine powder. S2: the extrudate is rounded to obtain pellets in the form of rods, dumbbells, or double balls. S3: the extrudate is rounded to obtain pellets in the form of spheres. S4: the extrudate after rounding becomes a large ball or an irregularly shaped mass.

**Table 3 pharmaceuticals-16-01505-t003:** Physical properties of the extrudates with pure MCC prescriptions (*n* = 3, x ± s).

No.	Model Drug	Water Added (%)	Ha(g)	Ad (g.s)	Sp	Co	Ch	Re
1	JRS101	100	19,869.63 ± 396.54	−10.88 ± 2.80	0.45 ± 0.02	0.35 ± 0.01	3168.22 ± 207.63	0.13 ± 0.00
2	JRS101	110	17,474.50 ± 113.67	−16.15 ± 4.58	0.57 ± 0.02	0.34 ± 0.01	3326.95 ± 9.17	0.12 ± 0.00
3	JRS101	120	18,528.92 ± 272.93	−19.61 ± 1.56	0.58 ± 0.06	0.35 ± 0.02	3699.10 ± 402.50	0.12 ± 0.00
4	JRS101	130	14,663.68 ± 532.13	−46.31 ± 9.50	0.90 ± 0.09	0.30 ± 0.01	3913.71 ± 209.05	0.11 ± 0.00
5	Avicel101	100	19,079.02 ± 191.21	−11.97 ± 19.00	0.38 ± 0.10	0.34 ± 0.00	2446.28 ± 275.97	0.12 ± 0.00
6	Avicel101	110	18,722.43 ± 592.77	−15.81 ± 1.01	0.43 ± 0.04	0.37 ± 0.01	2927.12 ± 356.88	0.12 ± 0.00
7	Avicel101	120	18,018.31 ± 438.75	−10.32 ± 1.39	0.53 ± 0.12	0.33 ± 0.03	3127.08 ± 315.07	0.12 ± 0.01
8	Avicel101	130	15,596.65 ± 124.50	−20.29 ± 1.43	0.71 ± 0.06	0.32 ± 0.01	3516.64 ± 149.55	0.12 ± 0.00
9	MT101	100	16,467.22 ± 261.69	−9.41 ± 14.09	0.71 ± 0.00	0.32 ± 0.01	3713.14 ± 112.44	0.12 ± 0.00
10	MT101	110	17,082.94 ± 666.16	−11.46 ± 1.94	0.66 ± 0.01	0.32 ± 0.01	3624.37 ± 226.95	0.12 ± 0.00
11	MT101	120	14,428.62 ± 429.36	−12.34 ± 31.11	0.76 ± 0.27	0.30 ± 0.00	3310.52 ± 1551.97	0.11 ± 0.00
12	MT101	130	15,929.77 ± 605.83	−19.00 ± 37.73	0.72 ± 0.31	0.32 ± 0.01	3675.05 ± 888.95	0.12 ± 0.01
13	Oricial101	100	20,268.09 ± 923.39	−8.87 ± 0.73	0.55 ± 0.08	0.35 ± 0.01	3902.75 ± 675.46	0.14 ± 0.01
14	Oricial101	110	17,984.93 ± 535.22	−8.90 ± 3.15	0.40 ± 0.04	0.33 ± 0.01	2385.07 ± 342.51	0.11 ± 0.00
15	Oricial101	120	17,903.57 ± 341.95	−16.03 ± 6.99	0.71 ± 0.09	0.34 ± 0.01	4308.82 ± 422.75	0.13 ± 0.00
16	Oricial101	130	15,745.03 ± 497.76	−19.70 ± 7.99	0.67 ± 0.02	0.32 ± 0.01	3369.69 ± 282.72	0.12 ± 0.00
17	JRS102	100	19,806.77 ± 526.25	−7.01 ± 2.47	0.45 ± 0.02	0.33 ± 0.01	2926.46 ± 171.67	0.12 ± 0.00
18	JRS102	110	18,087.71 ± 781.22	−13.20 ± 0.05	0.73 ± 0.00	0.31 ± 0.01	4148.26 ± 757.56	0.12 ± 0.01
19	JRS102	120	16,815.14 ± 623.08	−23.25 ± 5.51	0.63 ± 0.05	0.33 ± 0.02	3499.78 ± 312.13	0.12 ± 0.00
20	JRS102	130	15,151.74 ± 404.74	−22.31 ± 5.02	0.74 ± 0.08	0.29 ± 0.01	3283.08 ± 362.55	0.11 ± 0.00
21	Avicel113	100	19,921.93 ± 816.28	−10.95 ± 5.08	0.53 ± 0.11	0.38 ± 0.04	4135.74 ± 1531.56	0.14 ± 0.03
22	Avicel113	110	19,271.23 ± 33.91	−10.60 ± 2.78	0.49 ± 0.06	0.36 ± 0.01	3355.25 ± 433.77	0.13 ± 0.01
23	Avicel113	120	16,536.09 ± 702.53	−30.70 ± 7.28	0.86 ± 0.17	0.30 ± 0.02	4244.80 ± 641.24	0.11 ± 0.01
24	Avicel113	130	15,831.49 ± 622.43	−22.52 ± 6.71	0.80 ± 0.13	0.31 ± 0.01	3930.65 ± 638.37	0.12 ± 0.00
25	SH-101	100	16,260.23 ± 186.62	−6.15 ± 4.27	0.25 ± 0.02	0.33 ± 0.01	1316.27 ± 137.87	0.13 ± 0.01
26	SH-101	110	15,087.91 ± 349.23	−14.25 ± 53.06	0.37 ± 0.18	0.32 ± 0.01	1755.53 ± 841.45	0.12 ± 0.00
27	SH-101	120	13,798.89 ± 777.69	−13.85 ± 2.95	0.28 ± 0.05	0.30 ± 0.01	1169.06 ± 183.11	0.12 ± 0.00
28	SH-101	130	13,427.07 ± 1121.69	−17.08 ± 47.09	0.24 ± 0.04	0.30 ± 0.01	986.47 ± 121.97	0.12 ± 0.00

Notes—Adhesiveness (Ad) (g.s) is the negative area of the curve from the first compression force of zero to the second compression and represents the size of the adhesion between particles and particle surfaces, including the adhesion between the same particle surface and different particle surfaces. Springiness (Sp) is the ability of the extrudate to finally recover when the applied force disappears. Cohesiveness (Co) is the mutual attraction between the same particles in the same substance. Chewiness (Ch) represents the ease of the extrudate being cut off. Resilience (Re) is the instant reversibility of the extrusion.

**Table 4 pharmaceuticals-16-01505-t004:** LSD comparison of the textural properties and surface roughness of single TCM prescriptions and pure MCC prescriptions for obtaining spherical pellets (mean ± S.D.).

The Type of Extrusion-Round	Ha (g)	Ad (g.s)	Sp *	Co *	Ch *	Re	R *
E3-S3_TCM_	17,362 ± 2667	−31.966 ± 44.404	0.537 ± 0.173	0.349 ± 0.032	3150 ± 728	0.111 ± 0.012	1.30 ± 0.040
E2-S3_MCC_	16,607 ± 18	−18.486 ± 5.998	0.686 ± 0.098	0.319 ± 0.015	3589 ± 355	0.117 ± 0.005	1.16 ± 0.030

* The significance level of mean difference is 0.01.

**Table 5 pharmaceuticals-16-01505-t005:** Spearman’s correlation analysis between the texture characteristic parameters and surface roughness values of the extrudates in the prescriptions containing single TCM extracts.

The Correlation with R	Ha (g)	∣Ad∣ (g.s)	Sp	Co	Ch	Re
Correlation coefficient	0.128 *	−0.311 **	−0.259 **	0.298 **	−0.112	0.213 **
*p*	0.030	0.000	0.000	0.000	0.058	0.000

** The correlation is significant when the confidence level (bilateral) is 0.01. * The correlation is significant when the confidence level (bilateral) is 0.05.

**Table 6 pharmaceuticals-16-01505-t006:** Spearman’s correlation analysis between the texture property parameters and surface roughness values for the pure MCC formulations.

The Correlation with R	Ha (g)	∣Ad∣ (g.s)	Sp	Co	Ch	Re
Correlation coefficient	−0.158	−0.076	−0.108	−0.060	−0.137	0.100
*p*	0.421	0.700	0.586	0.763	0.485	0.310

**Table 7 pharmaceuticals-16-01505-t007:** ANOVA results of the regression analysis of the texture characteristic parameters and surface roughness R of the TCM prescriptions.

Model		Sum of Squares	df	Mean Square	F	Sig.
1	Regression	0.201	1	0.201	38.454	0.000 ^a^
	Residual error	1.494	286	0.005		
	Total	1.695	287			
2	Regression	0.238	2	0.119	23.320	0.000 ^b^
	Residual error	1.456	285	0.005		
	Total	1.695	287			
3	Regression	0.347	3	3.116	24.356	0.000 ^c^
	Residual error	1.348	284	0.005		
	Total	1.695	287			

^a^. Prediction variable: (constant), X3; ^b^. prediction variables: (constant), X3, X1; ^c^. prediction variables: (constant), X3, X1, and X4.

**Table 8 pharmaceuticals-16-01505-t008:** Coefficient of regression analysis of the texture characteristic parameters and surface roughness R of the TCM prescriptions.

Model		Nonstandardized Coefficient		Standard Coefficient	t	Sig.	Collinear Statistics	
		B	Standard Error	Trial Version			Tolerance	VIF
1	(constant)	1.329	0.013		105.562	0.000		
	X3	−0.125	0.020	−0.344	−6.201	0.000	1.000	1.000
2	(constant)	1.427	0.038		37.154	0.000		
	X3	−0.173	0.027	−0.478	−6.476	0.000	0.555	1.803
	X1	−4.040 × 10^−6^	0.000	−0.200	−2.708	0.007	0.555	1.803
3	(constant)	1.203	0.060		20.149	0.000		
	X3	−0.104	0.030	−0.287	−3.523	0.000	0.422	2.370
	X1	−9.697 × 10^−6^	0.000	−0.479	−5.208	0.000	0.331	3.024
	X4	0.829	0.173	0.500	4.780	0.000	0.256	3.899

## Data Availability

Data is contained within the article and Appendix A.

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
