# Peer review of "Influence of Extrudate-Based Textural Properties on Pellet Molding Quality"

_pharmaceuticals, 2023, doi:10.3390/ph16101505_

Round 1

Reviewer 1 Report

There are many information is missing, for example

  1. What is MTR?
  2. It is not clear what is the r value and how to calculate it.
  3. Why seven brands of MCC are used? Some of these MCC samples have same particle size. Did changing the vendor play a role in the extrusion evaluation? Authors need to include the MCC specifications for the 7 kinds.
  4. It was not clear what kind of MCC was used for the herbal extract pellets.
  5. The units of some of the parameters are missing in the text as well as in tables and Figures.
  6. There are some typos and incomplete sentences throughout the text. 
  1. There are some typos and incomplete sentences throughout the text. 

Reviewer 2 Report

The current manuscript entitled “Influence if extrudate-based textural properties on pellet molding quality” is interesting. The manuscript needs to be improved in few areas for better understanding. Please address the below comments:

1.     Please improve the quality of the abstract for more in depth and overall understanding.

2.     Please capture the scope of current work in the last section of the introduction.

3.     Authors are requested to improve the content in the discussion section of the manuscript. The results are superficially discussed, more in justification and supporting literature is needed.

4.     Please provide the full form of the abbreviations when used for the first time within the text. Cross-check the entire manuscript.

5.     The conclusion needs to be updated. Focus the main findings of the reseach.

6.     Cross check the references for missing information.

Minor English corrections are required. 

Reviewer 3 Report

The topic of this article is interesting, the authors presenting the preparation, characterization and evaluation the influence of extrudate-based textural properties on the quality of various pellets obtained under the same extrusion-spheronization conditions.

Concerning the content of the manuscript, I have the following remarks:

The abstract should provide a concise summary of the main results and objectives of the study. If possible, include some important details, such as key experimental findings or research methods employed. Please add more detailed information regarding the research objectives and specific applications of the different types of pellets obtained.

At the end of the introduction the objective of the work and how it was carried out should be briefly mentioned.

In the discussion section, the authors need to highlight the novelty of their researches and, to develop argumentation in depth based on the current understanding and the findings of the results obtained, presenting the potential, the weakness and future research direction, among others. Authors should try to explain the theoretical implication as well as the translational application of their research.

Conclusions need to be specified and much more synthetized. Now it looks like a very lengthy description of the results obtained.

I have also identified some important flaws:

-    the authors should improve the quality of figures 4 and 5, and to modify the font size for better clarity;

-    the abbreviations MCC, MTR, R (figure 2), and others should be explained;

-    the authors should mandatory upgrade the references;

-    in the references section please use the internationally recognized abbreviation of the journal;

-    at the references the authors should provide the DOI of the all the articles;

spelling check of the text is required;

 English including grammar, style and syntax, should be improved through the professional help from English Editing Company for Scientific Writings.

Reviewer 4 Report

General comments

The work appears to be carried out rigorously, but the nature of the experimental data must be explained in greater detail for the reader (or at least me) to grasp the value of the work. At present, I have trouble to assess the quality of the contribution.

Specific comments:

Abstract: The manuscript remains vague with “traditional Chinese medicine formulations”. Why opt for such terms?

Near line 100: Please clearly state what quality index is used.

Table 1: Identify R and include units in table caption.

Table 1 and Figure 1: explain extrusion and rounding results (E1, E2…). You conclude that “10 prescriptions could not be smoothly extruded” I do not understand this from your manuscript.

Figure 2: R values are not explained. 

Error bars must be included in the results. This being said, the appropriate number of decimals must be used in Table 4. 

ok

Round 2

Reviewer 1 Report

The authors addressed all my comments. 

Thanks,

Reviewer 3 Report

The authors have significantly revised the manuscript addressing the concern raised. I consider it could be accepted for publication in this journal.

Minor editing of English language is required 

Reviewer 4 Report

The manuscript is improved. However, I do not believe such large tables should be resent in the main body of the manuscript. I suggest placing them in supplementary materials and finding a way to sum them up in the main body of the text.

No further comments.
